# One-Dimensional Simulation of Synergistic Desulfurization and Denitrification Processes for Electrostatic Precipitators Based on a Fluid-Chemical Reaction Hybrid Model

**Chao Zhang** 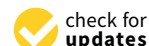 **and Lixin Yang ***

Beijing Key Laboratory of Flow and Heat Transfer of Phase Changing in Micro and Small Scale,
Beijing Jiaotong University, Beijing 100044, China; 16121371@bjtu.edu.cn
***** Correspondence: lxyang1@bjtu.edu.cn; Tel.: +86-010-5168-4329

**Abstract:** Non-thermal plasma (NTP) technologies can be used to treat a variety of gaseous pollutants, and extensive research has been carried out worldwide because of its high purification efficiency, low dependence on temperature, and other advantages. NO and $SO_2$ are the main gaseous pollutants in coal-fired flue gas. The plasma dynamics for desulfurization and denitrification is a hot topic in the field of NTP pollutant control technologies. In this paper, a one-dimensional fluid model for the simultaneous desulfurization and denitrification of flue gas by negative direct current (DC) corona discharge was established based on the traditional zero-dimensional chemical kinetic model. The simplified wire-cylindrical electrodes configuration and numerical simulation conditions are similar to the working process of electrostatic precipitators. The results obtained by the finite element method show that the removal efficiency of NO and $SO_2$ is remarkable in the region with a radius of less than one centimeter around the high-voltage electrode, and the effective purification area expands with the increase of the discharge voltage. There are different removal pathways for NO at different positions in the removal region, while the removal of $SO_2$ is mainly dependent on the oxidation by OH.

**Keywords:** one-dimensional fluid model; chemical kinetic model; non-thermal plasma; desulfurization; denitrification

## 1. Introduction

Non-thermal plasma (NTP) is characterized by its internal electron temperature, which is far higher than that of ions and neutral particles. The electron energy usually comes from the acceleration of electrons by an external electric field. Recent studies have shown that NTP has good performance for the removal of gaseous pollutants such as NO and $SO_2$ [1–3], and industrial demonstration applications have been carried out because of its compact structure and high energy-efficiency [4,5]. Gas discharge at atmospheric pressure is the most common form of NTP used for flue gas treatment, such as corona discharge, dielectric barrier discharge, and so on [6,7]. The realization of desulfurization and denitrification relies on the highly selective chemical reactions between pollutant molecules and the active particles in plasma, which come from the collisions between high-energy electrons and background gas molecules. According to the time scale, the flue gas purification process under gas discharge conditions can be divided into two stages [8]. The primary stage is composed of a series of inelastic collision reactions initiated by high-energy electrons, which includes ionization, dissociation, excitation, and other types with the time scale of about $10^{-8}$ s. A large number of ions and free radicals produced in this stage interact with each other in the secondary stage with the time scale of about

$10^{-3}$ s and produce more active substances, which are responsible for the conversion of harmful gas molecules into harmless or easily removable forms.

At present, the simulation of the desulfurization and denitrification processes by NTP is mainly focused on the numerical calculation of the microscopic chemical reaction mechanism through special programs. Li et al. [9] constructed a chemical kinetic model containing 48 elementary reactions for pulsed discharge. The concentration variation of various gaseous components with time in two gas systems of $NO/SO_2/N_2/O_2$ and $NO/SO_2/N_2/O_2/H_2O$ was calculated, and the effect of oxygen content on the removal efficiency of NO and $SO_2$ was analyzed. Teodoru et al. [10] summarized the reaction path of the removal of $NO_x$ under the condition of dielectric barrier discharge in the system of $NO_x/N_2/O_2/H_2O$, as well as the energy consumption and by-product formation with or without oxygen by solving a model consisting of 540 reactions. Yin et al. [11] calculated the time evolution of NO and its oxidation or reduction products in the weakly ionized gas of mixed NO, $N_2$, and $O_2$ at normal temperature and pressure. The main reactions and active species affecting the formation and consumption of NO were obtained by chemical sensitivity analysis, and the employed chemical mechanism involved 35 species and 225 reactions. The chemical kinetic model and its solution, which were involved in the above studies, are essentially zero-dimensional, that is, assuming that desulfurization and denitrification are carried out in NTP with uniform spatial distribution, and ignoring the spatial characteristics of particle transport and interaction in plasma.

The fluid model is widely used in the research of NTP, by which the physical and chemical characteristics simulation of NTP in one-dimensional [12], two-dimensional [13], and three-dimensional space scales [14] can be realized. In this paper, a one-dimensional hybrid model of negative DC corona discharge was established by coupling the fluid model with the traditional chemical kinetic model [15,16]. The emphasis is laid on the analysis of the spatiotemporal evolution of desulfurization and denitrification between the discharge electrode and grounding electrode, as well as the influence of discharge voltage and background gas composition on the discharge characteristics and removal efficiency.

## 2. Model Description

The fluid model of gas discharge mainly includes the transport equation of electron and electron energy, the transport equation of heavy particles, the Poisson equation, and so on, which can be used to solve the NTP process in a multi-dimensional space under the conditions of atmospheric pressure and complicated chemical reaction mechanisms. Therefore, it has an advantage among the mathematical models for NTP simulation. The calculation of electron transport coefficients and the rate coefficients of electron collision reactions are the prerequisites for solving the fluid model. In the NTP dynamics, the Boltzmann equation is usually used to constrain the time, space, and velocity distributions of electrons in partially ionized gases. The electron energy distribution function (EEDF) under specific gas composition can be obtained by solving the Boltzmann equation; then, the parameters required by the fluid model can be obtained.

### 2.1. Governing Equations

The transport equation of electron and electron energy in the fluid model is obtained from a two-order approximate solution of the Boltzmann Equation [17], as shown in Equations (1) and (2):

$$\frac{\partial n_e}{\partial t} + \vec{\nabla} \cdot \vec{\Gamma}_e = S_e \tag{1}$$

$$\frac{\partial (n_e \bar{\varepsilon})}{\partial t} + \vec{\nabla} \cdot \vec{\Gamma}_\varepsilon + \vec{E} \cdot \vec{\Gamma}_\varepsilon = S_\varepsilon \tag{2}$$

where $n_e$ is the electron density; $\vec{\Gamma}_e$ is the electron flux; $S_e$ is the electron source term; $\bar{\varepsilon}$ is the mean electron energy; $\vec{\Gamma}_\varepsilon$ is the electron energy flux; $S_\varepsilon$ is the electron energy source term; and $\vec{E}$ is the vector of electric field intensity. The electron flux and electron energy flux are shown in Equations (3) and (4):

$$\vec{\Gamma}_e = -D_e \cdot \vec{\nabla} n_e - n_e \cdot \mu_e \cdot \vec{E} \tag{3}$$

$$\vec{\Gamma}_\varepsilon = -D_\varepsilon \cdot \vec{\nabla}(n_e \bar{\varepsilon}) - (n_e \bar{\varepsilon})\mu_\varepsilon \cdot \vec{E} \tag{4}$$

where $\mu_e$ is the electron mobility; $D_e$ is the electron diffusion coefficient; $\mu_\varepsilon$ is the electron energy mobility; and $D_\varepsilon$ is the electron energy diffusion coefficient. These four parameters are shown in Equations (5)–(8):

$$\mu_e = -\frac{\gamma}{3N} \int_0^\infty \frac{\varepsilon}{\sigma_m + \overline{v}_i / N\gamma \varepsilon^{1/2}} \frac{\partial f_0}{\partial \varepsilon} d\varepsilon \tag{5}$$

$$D_e = \frac{\gamma}{3N} \int_0^\infty \frac{\varepsilon}{\sigma_m + \overline{v}_i / N\gamma \varepsilon^{1/2}} f_0 d\varepsilon \tag{6}$$

$$\mu_\varepsilon = -\frac{\gamma}{3N\bar{\varepsilon}} \int_0^\infty \frac{\varepsilon^2}{\sigma_m + \overline{v}_i / N\gamma \varepsilon^{1/2}} \frac{\partial f_0}{\partial \varepsilon} d\varepsilon \tag{7}$$

$$D_\varepsilon = \frac{\gamma}{3N\bar{\varepsilon}} \int_0^\infty \frac{\varepsilon^2}{\sigma_m + \overline{v}_i / N\gamma \varepsilon^{1/2}} f_0 d\varepsilon \tag{8}$$

where $\gamma = (2e/m_e)^{1/2}$; $e$ is the amount of charge carried by a single electron; $m_e$ is the mass of a single electron; $N$ is the number density of heavy particles; $\varepsilon$ is the electron energy; $\sigma_m$ is the sum of electron collision cross-sections; $\overline{v}_i$ is the net generation frequency of electrons; and $f_0$ is the isotropic part of electron distribution function simplified by two-order approximation. The electron source term and electron energy source term mentioned above can be expressed in the form of Equations (9) and (10) [17]:

$$R_e = \sum_{j=1}^M x_j k_j N_n n_e \tag{9}$$

$$R_\varepsilon = \sum_{j=1}^P x_j k_j N_n n_e \Delta \varepsilon_j \tag{10}$$

where $M$ is the number of reactions that cause changes in the number of electrons; $P$ is the number of reactions that cause changes in electron energy; $x_j$ is the mole fraction of reactant collided with electrons in reaction $j$; $k_j$ is the rate coefficient of reaction $j$; $N_n$ is the number density of neutral particles; and $\Delta \varepsilon_j$ is the change of electron energy caused by reaction $j$.

Heavy particles represent the positive and negative ions as well as neutral particles in NTP. The transport equation of heavy particles involved in the fluid model is shown in Equation (11) [18]:

$$\frac{\partial n_k}{\partial t} + \left(\vec{u} \cdot \vec{\nabla}\right) n_k = R_k + \vec{\nabla} \cdot \vec{j}_k \tag{11}$$

where $n_k$ is the number density of heavy particle $k$; $\vec{u}$ is the vector of mean mass velocity; $R_k$ is the source term of heavy particle $k$; and $\vec{j}_k$ is the diffusion flux of heavy particle $k$, as shown in Equation (12):

$$\vec{j}_k = n_k \left( \sum_{i=1}^Q \widetilde{D}_{ki} \vec{d}_k - \frac{D_k^T}{N_A M_k n_k} \vec{\nabla} \ln T \right) \tag{12}$$

where $Q$ is the total number of species in the plasma system; $\widetilde{D}_{ki}$ is the Maxwell–Stefan diffusion coefficient; $\vec{d}_k$ is the diffusion driving force [19]; $D_k^T$ is the thermodynamic diffusion coefficient; $N_A$ is

the Avogadro constant; and $M_k$ is the molar mass. The source term of heavy particle $k$ can be expressed as the form of Equation (13):

$$R_k = N_A \sum_{j=1}^{W} v_{kj} r_j \tag{13}$$

where $W$ is the number of reactions leading to changes in the number of heavy particles; $v_{kj}$ is the stoichiometric matrix; and $r_j$ is the reaction rate of reaction $j$.

Finally, the space electric field is described by the Poisson Equation (14) [20]:

$$\vec{\nabla} \cdot \left( \vec{\nabla} \varphi \right) = -\frac{e}{\varepsilon_r \varepsilon_0} \left( \sum_p n_p - n_e - \sum_n n_n \right) \tag{14}$$

where $\varphi$ is the space electric potential; $\varepsilon_r$ is the relative dielectric constant; $\varepsilon_0$ is the dielectric constant of vacuum; $n_p$ is the number density of positively charged particles; and $n_n$ is the number density of negatively charged particles.

## 2.2. Electron Collision and Chemical Kinetic Model

The transport equations of electron and heavy particles involve the source terms of electron and heavy particles, respectively. Equations (9) and (13) show that the formation and annihilation of electrons and heavy particles, as well as the mutual transformation between different species, can be attributed to the complex chemical reactions in NTP. The essence of chemical reactions is the collisions between different particles. As for the collision process between specific particles, the difference of momentum exchange, energy exchange, and charge transfer will be caused by the difference among the kinetic energy of the particles. According to whether the total kinetic energy of particles is conserved before and after the collision, the collisions in plasma can be divided into elastic collisions and inelastic collisions. Elastic collisions follow the conservation of total kinetic energy, and there is no transformation of particle species, because they do not cause changes in the internal energy of particles. Meanwhile, inelastic collisions cause a loss of kinetic energy and changes in the internal energy of particles, which is usually accompanied by the generation of new species or the annihilation of the original species.

The chemical reaction mechanism is an important part of the fluid model. Based on the main background gas components of coal-fired flue gas, the electron collision kinetic model and chemical kinetic model for the desulfurization and denitrification of flue gas by corona discharge were constructed in this paper.

### 2.2.1. Electron Collision Kinetic Model

The streamer theory of gas discharge shows that the generation of electrons and ions in plasma is mainly dependent on the ionization of gas molecules or atoms by the collisions with high-energy electrons under an external electric field, and the resulting electron avalanche [21]. In fact, the collision reactions between electrons and gas molecules or atoms include elastic collision, ionization, attachment, dissociation, excitation, and so on, which produce positive and negative ions, primary free radicals, excited particles, and other species. The formation of these species will have a decisive impact on the subsequent chemical reaction pathways, such as the generation of secondary free radicals and the removal of pollutants.

As NO, $SO_2$, and other gaseous pollutants in coal-fired flue gas are trace components relative to background gases such as $O_2$, $N_2$, $H_2O$, and $CO_2$, the composition of pollutants can be neglected when considering the collisions between high-energy electrons and gaseous molecules or atoms. In addition, for each kind of background gas, it is necessary to consider all kinds of collision reactions that may occur due to different electron energy, especially the reactions involving active particles (such as O, $O(^1D)$, $O_2(a^1\Delta_g)$, $N_2(A)$, OH, etc.) related to flue gas purification. Based on the above considerations, a

kinetic model of electron collision consisting of 64 elementary reactions was constructed, as shown in Appendix A. The reaction mechanism comes from the open-access website LXcat [22] and a series of sub-databases [23,24].

The electron collision cross-section is an important concept in plasma dynamics, whose physical meaning is the probability of collision between electrons and heavy particles in a specific electron collision reaction. Each electron collision reaction has its corresponding collision cross-section data, which is usually a function of the incident electron energy. The electron collision cross-section data of reaction R1-R64 is also obtained from the website LXcat [22].

### 2.2.2. Chemical Kinetic Model

In real flue gas, the number of reactions and the types of particles involved in the processes of desulfurization and denitrification by corona discharge are massive, and the rate coefficients of many reactions are unknown. It is also difficult to select the most important part from the various reaction mechanisms proposed by different scholars and assemble a complete chemical kinetic model. Therefore, the chemical kinetic model that is constructed in this paper takes the data from a single literature that was selected as the main body, and is supplemented by a small amount of data from other literature, so as to reflect the main pathways of NO and $SO_2$ removal, and include the typical gaseous species of flue gas discharge. Based on the reaction mechanism in Reference [15] and the ion-related chemical reactions in Reference [16], a chemical kinetic model consisting of 117 elementary reactions was constructed, as shown in Appendix B.

### 2.2.3. Collision Reactions at the Electrode Surface

Positive and negative ions colliding with the electrode surface will become neutral heavy particles, and secondary electrons will be excited by positive ions bombarding the cathode surface. In addition, the excited particles will lose energy, and return to the ground state after collisions with the electrode surface [25].

### 2.3. Boundary Conditions

As mentioned above, there are many collision reactions occurring at the discharge and grounding electrode surface during gas discharge. Therefore, the boundary conditions at the electrode surface are one of the boundary conditions that need to be dealt with in the governing equations of the fluid model. In addition, the electrode also acts as the macroscopic boundary of the discharge gas, and the boundary conditions at the edge of discharge gas also need to be considered.

The physical and chemical behavior of electrons at the electrode surface includes reflection and secondary electron emissions. Therefore, the electrode boundary condition of the electron transport equation can be expressed as the form of Equation (15) [26]:

$$\overrightarrow{\Gamma}_e \cdot \overrightarrow{n} = \frac{1-r_e}{1+r_e}\left[-(2a_e-1)n_e\mu_e\overrightarrow{E}\cdot\overrightarrow{n} + \frac{1}{2}v_{th,e}n_e - \frac{1}{2}v_{th,e}n_\gamma\right] - \frac{2}{1+r_e}(1-a_e)\sum_p \gamma_p\overrightarrow{\Gamma}_p\cdot\overrightarrow{n} \tag{15}$$

$$v_{th,e} = \sqrt{\frac{8k_B T_e}{\pi m_e}} \tag{16}$$

$$n_\gamma = (1-a_e)\frac{\sum\limits_p \gamma_p\overrightarrow{\Gamma}_p\cdot\overrightarrow{n}}{\mu_e\overrightarrow{E}\cdot\overrightarrow{n}} \tag{17}$$

where the subscripts $e$ and $p$ represent the electrons and positive ions, respectively; $\overrightarrow{n}$ is the normal vector perpendicular to the electrode surface; $r_e$ is the reflection coefficient of the electrode surface to the electrons; $v_{th,e}$ is the electron thermal velocity; $n_\gamma$ is the electron density annihilated at the

electrode surface; $\gamma_p$ is the secondary electron emission coefficient at the cathode surface by positive ion bombardment; and $k_B$ is the Boltzmann's constant. The value of $a_e$ depends on the direction of electron motion. When the electron moves toward the electrode, the value of $a_e$ is 1; otherwise, it is 0.

The electrode boundary condition of the electron energy transport equation can be expressed in the form of Equation (18) [27]:

$$\overrightarrow{\Gamma}_\varepsilon \cdot \overrightarrow{n} = \frac{5}{3}\left(\frac{1}{4}v_{th,e}\bar{\varepsilon}n_e - \bar{\varepsilon}_p\frac{2}{1+r_e}(1-a_e)\sum_p\gamma_p\overrightarrow{\Gamma}_p \cdot \overrightarrow{n}\right) \tag{18}$$

where $\bar{\varepsilon}_p$ is the fixed initial energy of secondary electrons. The gas boundary conditions of the electron and electron energy transport equations can be expressed in the form of Equations (19) and (20):

$$-\overrightarrow{n} \cdot \overrightarrow{\Gamma}_e = 0 \tag{19}$$

$$-\overrightarrow{n} \cdot \overrightarrow{\Gamma}_\varepsilon = 0 \tag{20}$$

The positive and negative ions are transformed into neutral particles at the electrode surface, and the excited particles return to their ground state. Therefore, the electrode boundary condition of the transport equation of heavy particles can be expressed in the form of Equation (21):

$$\overrightarrow{\Gamma}_k \cdot \overrightarrow{n} = \frac{\gamma_k}{4}\sqrt{\frac{8k_BT_k}{\pi m_k}}n_k \tag{21}$$

where $\gamma_k$ is the collision reaction rate coefficient of heavy particle $k$ at the electrode surface. The gas boundary condition of the transport equation of heavy particles can be expressed in the form of Equation (22):

$$-\overrightarrow{n} \cdot \overrightarrow{\Gamma}_k = 0 \tag{22}$$

The gas boundary condition of Poisson equation can be expressed in the form of Equation (23) [25]:

$$\overrightarrow{n} \cdot \left(\varepsilon\overrightarrow{\nabla}\varphi\right) = 0 \tag{23}$$

*2.4. Physical Model*

The emphasis of the numerical simulation in this paper is to analyze the formation, transport, and interaction of charged and neutral particles in the processes of desulfurization and denitrification of flue gas by negative DC corona discharge, as well as the micromechanisms of the temporal and spatial evolution of flue gas discharge and pollutant removal between the discharge electrodes and grounding electrodes. In order to reduce the difficulty of numerical solution and the amount of calculation, the influence of the macroscopic electrode structure on the NTP process was selectively neglected. Therefore, the physical model was simplified, and the coaxial wire-cylindrical electrodes configuration that is commonly used in gas discharge was adopted. We assume that the NTP is uniform along the axis of the cylindrical grounding electrode, and only the radial inhomogeneity of discharge characteristics is considered. The simplified physical model is one-dimensional axisymmetric, and the solution domain is a one-dimensional geometric structure, as shown in Figure 1. The radius of the discharge electrode is 0.15 mm, and the distance from the center of the cross-section of the discharge electrode to the inner surface of the grounding electrode is 10 cm.

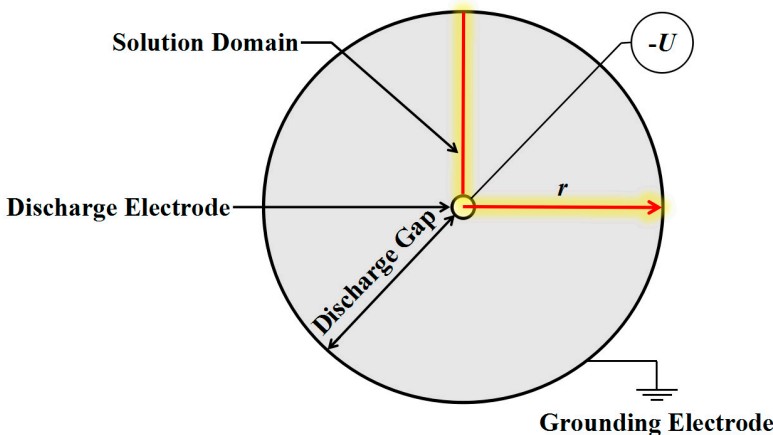

**Figure 1.** One-dimensional solution domain in the simplified physical model.

*2.5. Mesh Generation and Numerical Solution*

The evolution of NTP belongs to the category of multi-physics coupling, which is mathematically described as a system of partial differential equations. Therefore, the solution of the fluid model of desulfurization and the denitrification of flue gas by negative DC corona discharge can be reduced to the solution of partial differential equations. At present, the finite element method has become the mainstream in many engineering methods for solving partial differential equations.

In this paper, the finite element analysis software COMSOL Multiphysics was used to construct the numerical model, solve the finite element problems, and post-process the results. COMSOL Multiphysics has the Plasma Module for the simulation of NTP, which includes the Boltzmann equation, Two-Term Approximation interface, the Drift Diffusion interface, the Heavy Species Transport interface, and the Electrostatics interface. The Boltzmann equation and Two-Term Approximation interface were used to solve the governing equations of the EEDF. On the basis of the obtained EEDF, various transport coefficients of NTP were output and invoked by the solver. The electron collision kinetic model and collision cross-section data should be added to the interface. The Drift Diffusion interface was used to solve the transport equations of the electron and electron energy based on the Drift Diffusion Approximation, which is effective when the mean free path of electrons is much smaller than the macroscopic size of the discharge space. The Heavy Species Transport interface is used to solve the mass conservation equations of all of the non-electronic components, and the Electrostatics interface is used to solve the Poisson equation of the electrostatic field.

Mesh generation in the solution domain is an important step in the process of solving the fluid model with the finite element method. In the negative DC corona discharge, the characteristic parameters of NTP usually show a large gradient in the sheath region near electrodes, and their changes are more gentle in other areas. Therefore, the mesh near the electrodes was refined to improve the accuracy of the calculation results. The mesh element size is axisymmetrically distributed in the solution domain, and gradually increases in a geometric sequence from the surface of two electrodes to the midpoint of the solution domain, as shown in Figure 2. The total number of mesh elements is 3000, and the ratio of the mesh element size is 100.

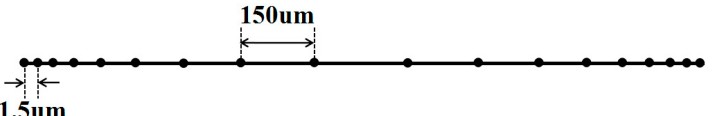

**Figure 2.** Mesh element distribution in a one-dimensional solution domain.

Some appropriate numerical methods were adopted to make the calculation easier in the initial stage of numerical solution without affecting the results. For example, the negative DC voltage

$U_0$ applied to the discharge electrode was described as the form of a step function, as shown in Equation (24):

$$U = U_0 \tan h \left( \frac{t}{\tau} \right) \tag{24}$$

where the constant $\tau$ was set to $10^{-5}$ s. In addition, the initial electron density of the model was set to $10^4$ cm$^{-3}$; the secondary electron emission coefficient was set to 0.05; and the mean energy of the secondary electrons was set to 4 eV.

## 3. Results and Discussion

In order to simulate the real coal-fired flue gas, the discharge background gas in the model is composed of $O_2$, $N_2$, $H_2O$, and $CO_2$, with a volume ratio of 0.033:0.741:0.083:0.143 [15]. The distribution of the reduced electric field, electron energy, and electron density will be discussed based on the calculation results. In addition, the reaction rates of the main collision reactions, the temporal and spatial evolution of NO, $SO_2$, and the active species, as well as the effects of discharge voltage on various parameters will be analyzed.

### 3.1. Effect of Discharge Voltage

Transient calculation was performed based on the one-dimensional fluid model and terminated to the time step of 0.83 s. The discharge gas pressure was set to one atm; the temperature was set to 300 K; and the initial number densities of NO and $SO_2$ were set to $2.4 \times 10^{15}$ cm$^{-3}$ (NO concentration is about 100 ppm, $SO_2$ concentration is about 200 ppm).

Figure 3 shows the distribution of the reduced electric field in the solution domain at the last time step under the discharge voltages of $-25$ KV, $-35$ KV, and $-45$ KV. Figure 3a shows that the reduced electric field decreases sharply from the edge of the discharge electrode, forming a narrow high electric field region near the discharge electrode and keeping a low level in the other areas. Within the distance of 1 mm from the discharge electrode, the reduced electric field is reduced by about one order of magnitude. The distribution of the reduced electric field in the high electric field region is quite similar under different voltages, and it shows a relatively large difference in other areas, as shown in Figure 3b.

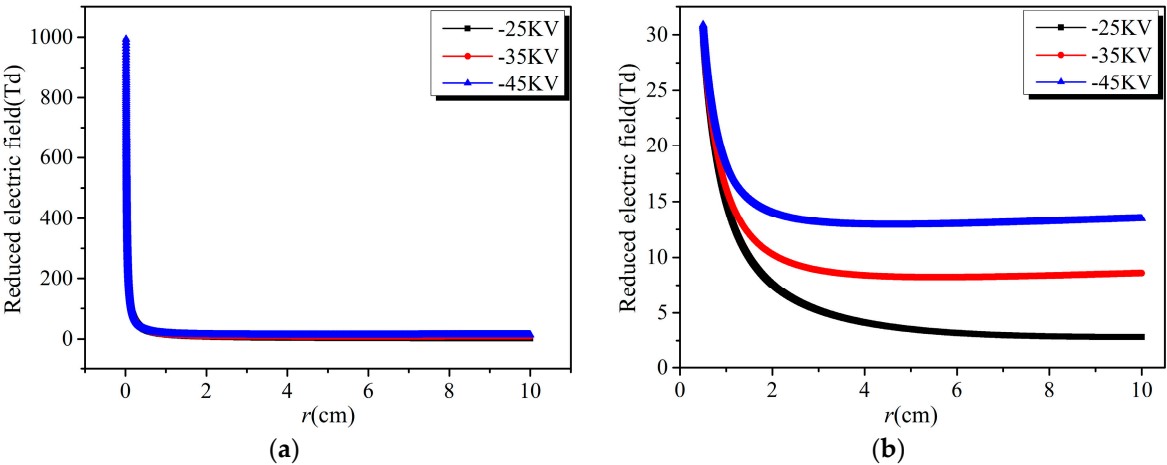

**Figure 3.** Effect of discharge voltage on reduced electric field: (**a**) Distribution of reduced electric field in the entire solution domain; (**b**) Distribution of reduced electric field in the low electric field region.

Figure 4 shows the distribution of electron energy in the solution domain at the last time step. The reduced electric field is an important parameter to synthetically measure the electric field force exerted on electrons and the mean free path of electrons in plasma. Under the same gas composition, the reduced electric field will directly determine the energy of free electrons. It can be seen that the distribution of electron energy and reduced electric field have the same trend, which shows the

dependence between them. The high electric field region is also the high electron energy region. The change of electron energy with discharge voltage in the low electron energy region is relatively obvious.

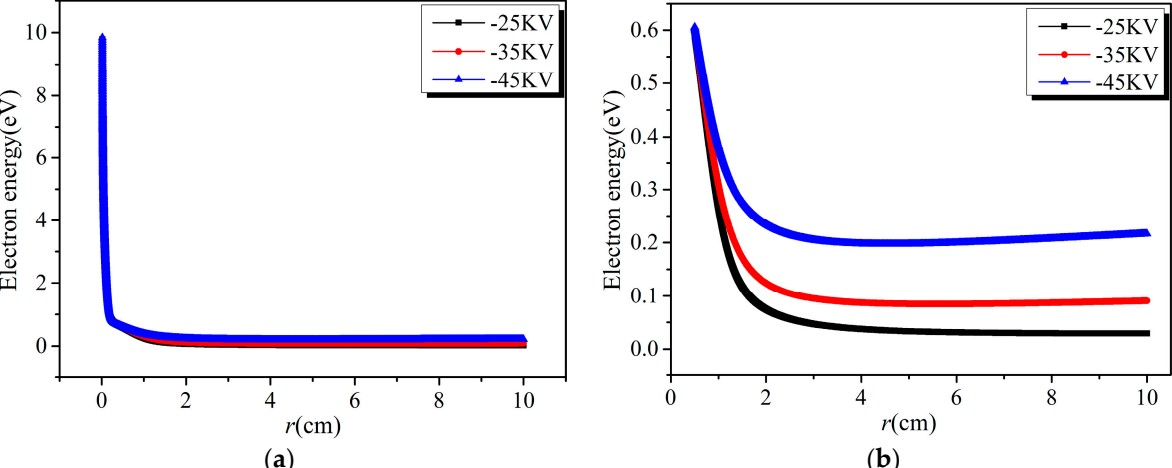

**Figure 4.** Effect of discharge voltage on electron energy: (**a**) Distribution of electron energy in the entire solution domain; (**b**) Distribution of electron energy in low electron energy region.

Figure 5 shows the distribution of the reaction rates of four electron collision reactions that have important effects on the generation of active particles at the time step of $2.9836 \times 10^{-5}$ s under the discharge voltages of $-25$ KV, $-35$ KV, and $-45$ KV. It can be seen that the four kinds of reaction rates are higher in the high electric field region that is about one mm away from the cathode, and tend to zero gradually outside this region. The high electric field region is also the main region of electron avalanche, where the high-energy electron density is relatively high, which leads to the higher reaction rates of the electron collision reactions. Figure 5 also shows that with the increase of discharge voltage, the collision reaction rates increase obviously in the high electric field region. Moreover, there is a peak value in the spatial distribution of the four reaction rates in the high electric field region, which is determined by the distribution of electron density and electron energy. From Figure 6e, it can be seen that the peak value of electron density also exists in the high electric field region. Obviously, the collision reaction rates are higher in the region with higher electron density, and the peak position of a specific collision reaction rate is governed by the location of the electron energy threshold corresponding to the reaction.

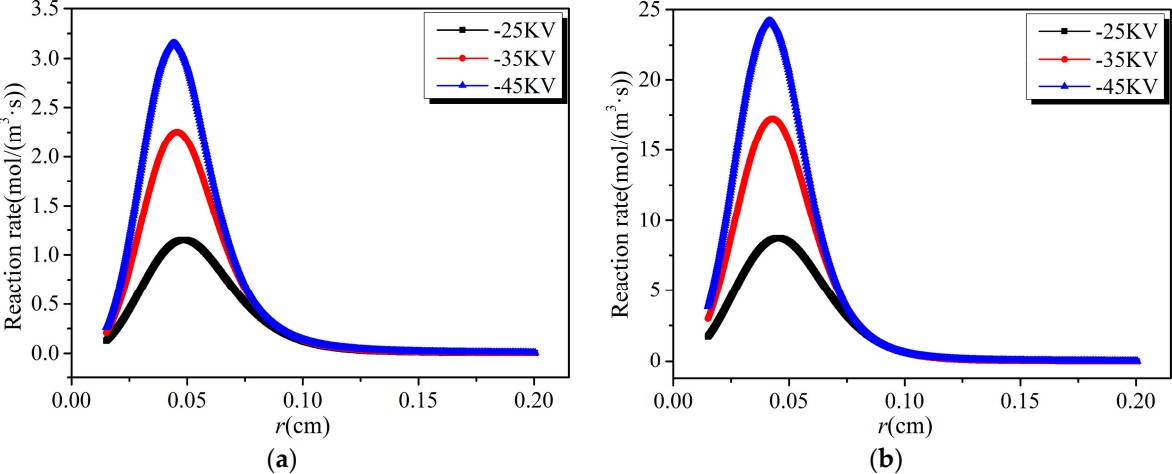

**Figure 5.** *Cont.*

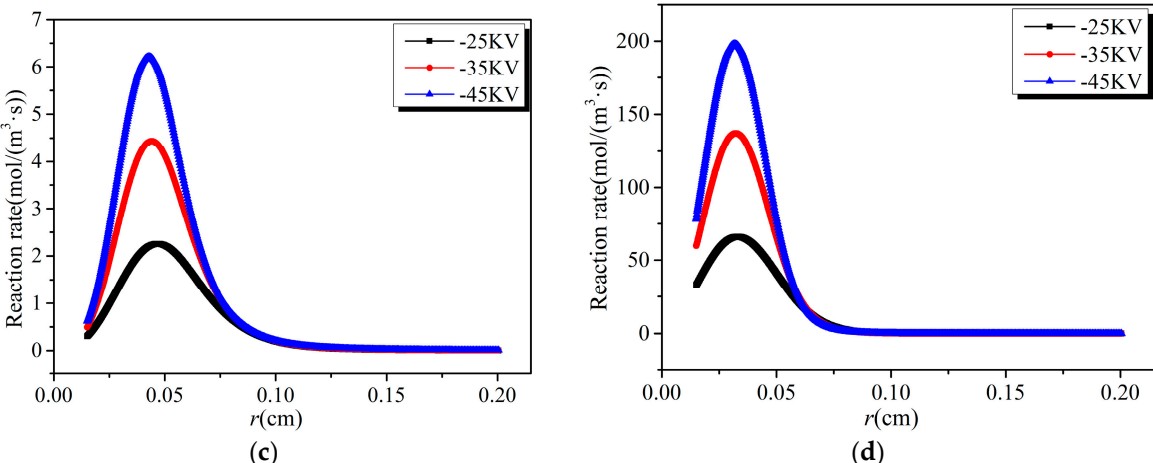

**Figure 5.** Effect of discharge voltage on reaction rates of electron collision reactions: (**a**) e + $O_2$ → e + O + O; (**b**) e + $O_2$ → e + O + O($^1$D); (**c**) e + $N_2$ → e + $N_2$(A); and (**d**) e + $N_2$ → e + N + N.

Figure 6 shows the distribution of the number densities of four kinds of active particles and electrons at the time step of $2.9836 \times 10^{-5}$ s. It can be seen that the distribution of the number densities of active particles between two electrodes has the same trend as that of the corresponding electron collision reaction rates, which shows the dependence between them. In Figure 6e, the electron avalanche in the high electric field region leads to the peak value of electron density in this region, which increases with the increase in the discharge voltage. During the discharge, the electrons generated in the high electric field region move toward the grounding electrode under the motivation of the external electric field, and the electron density near the grounding electrode increases gradually. When the discharge voltage rises, the electric field force on electrons, as well as the electron drift velocity in the low electric field region, both increase. Therefore, at the same time after discharge, the higher the discharge voltage, the higher the electron density near the grounding electrode, and the lower the electron density near the cathode, as shown in Figure 6e.

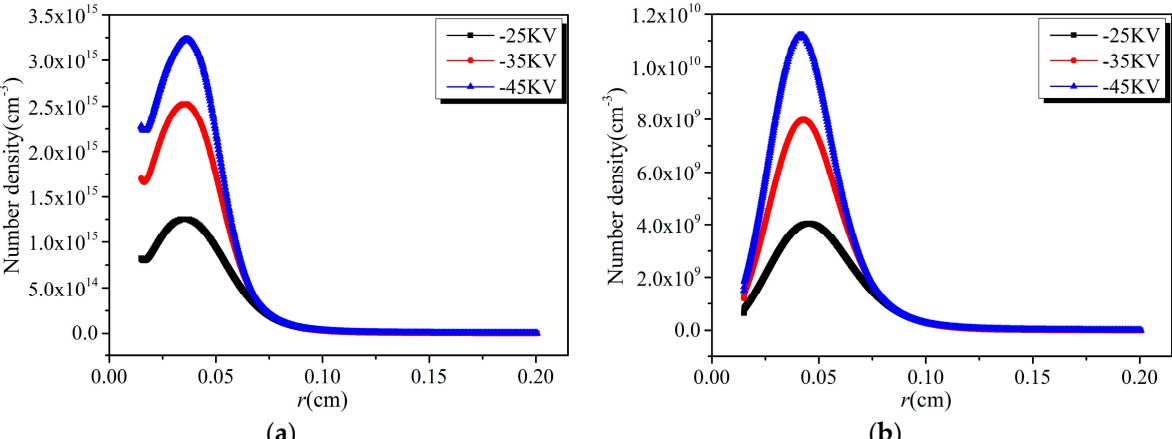

**Figure 6.** *Cont.*

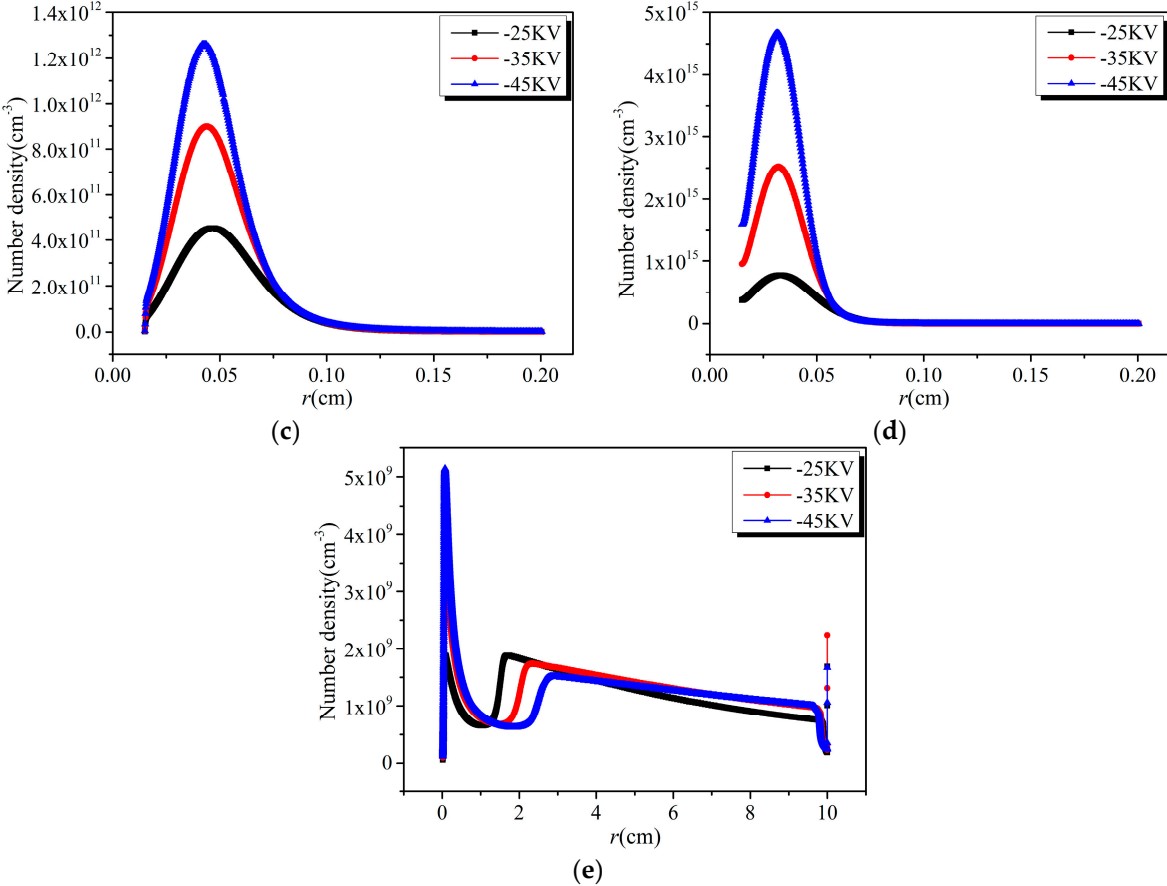

**Figure 6.** Effect of discharge voltage on the number densities of active particles and electrons: (**a**) O; (**b**) O($^1$D); (**c**) N$_2$(A); (**d**) N; and (**e**) *e*.

Figure 7 shows the temporal and spatial evolution of the number density of NO in the one-dimensional solution domain under the discharge voltages of −25 KV, −30 KV, −35 KV, −40 KV, −45 KV, and −50 KV. It can be seen that the effective removal area of NO increases with time, and the removal efficiency decreases with the increase of distance from the discharge electrode under the same discharge voltage. The above analysis shows that the high electric field region near the cathode is the main region of the distribution of active particles that are closely related to pollutant removal. Therefore, various NTP reactions, including pollutant removal reactions, perform actively in this region, resulting in a high NO removal efficiency. The directional migration and diffusion of the neutral and charged particles that are generated in the high electric field region lead to the widening of the distribution space of various species, so the effective removal area of NO increases with time. With the increase of discharge voltage, the effective removal area of NO, as well as the removal efficiency at the same position, both increase. This is because the number density of active particles increases with the increase of discharge voltage, so the diffusion of active particles is more active under the concentration gradient, resulting in the expansion of the removal area and the improvement of the removal efficiency. Under the calculation conditions in this paper, the effective removal area of NO by negative DC corona discharge is not more than one centimeter away from the discharge electrode.

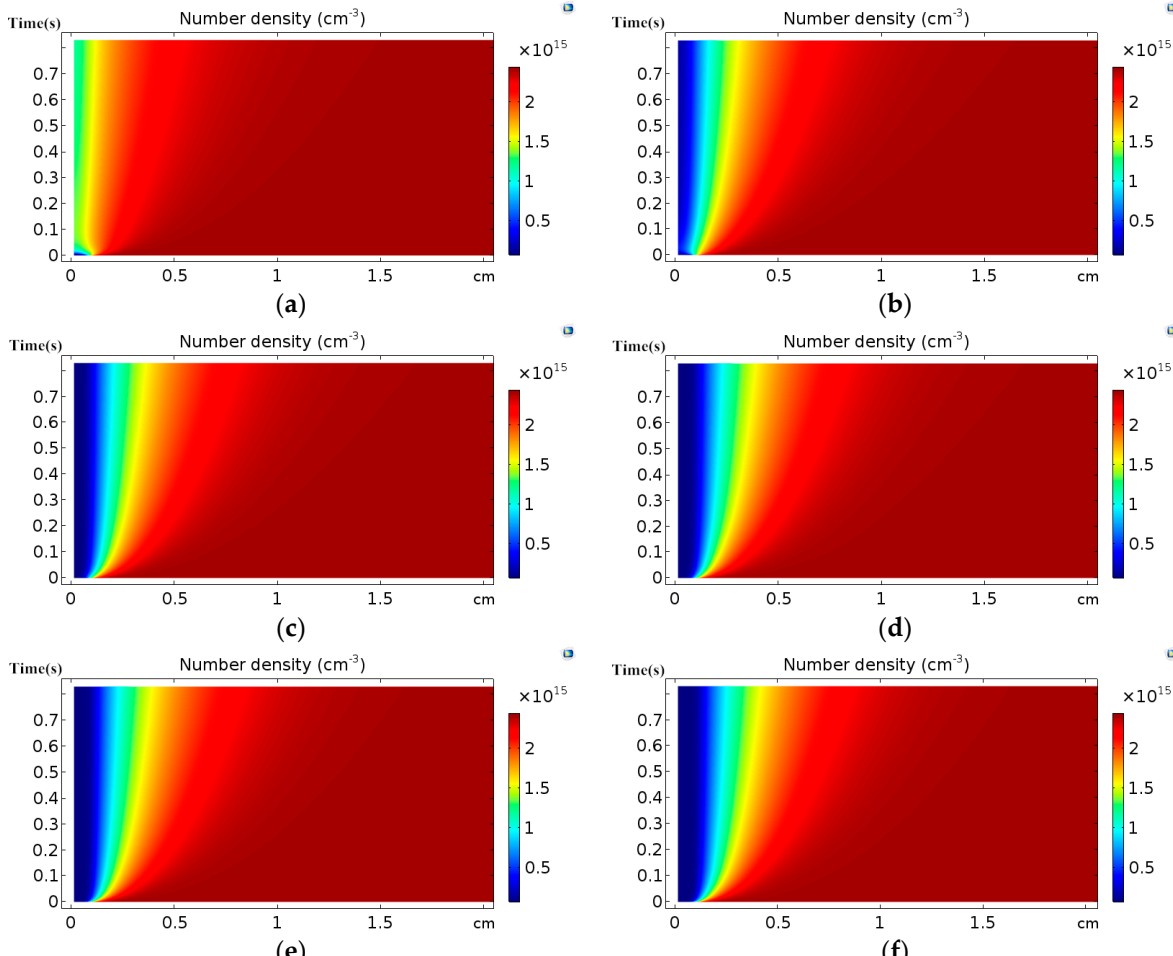

**Figure 7.** Effect of temporal and spatial evolution of the number density of NO: (**a**) −25 KV; (**b**) −30 KV; (**c**) −35 KV; (**d**) −40 KV; (**e**) −45 KV; and (**f**) −50 KV.

Figure 8 shows the temporal and spatial evolution of the number density of $SO_2$ under the discharge voltages of −25 KV, −30 KV, −35 KV, −40 KV, −45 KV, and −50 KV. It can be seen that the temporal and spatial evolution of $SO_2$ is similar to that of NO, which is determined by the common mechanism of the removal of gaseous pollutants by NTP. Under the calculation conditions in this paper, the effective removal area of $SO_2$ by negative DC corona discharge is not more than one centimeter away from the discharge electrode.

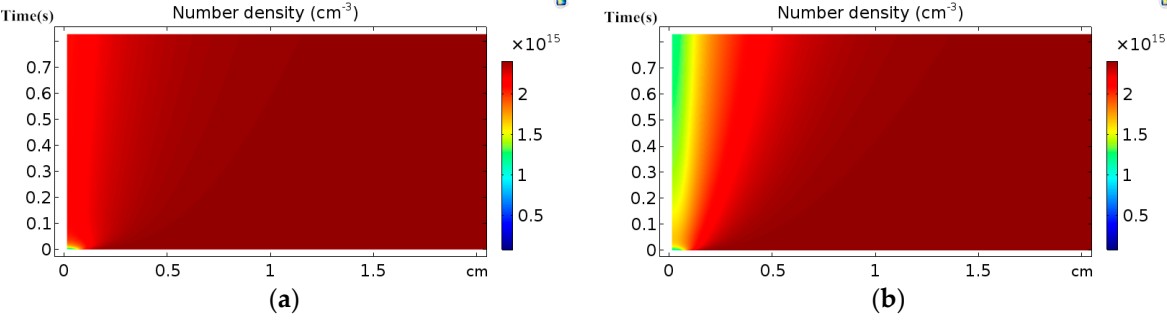

**Figure 8.** *Cont.*

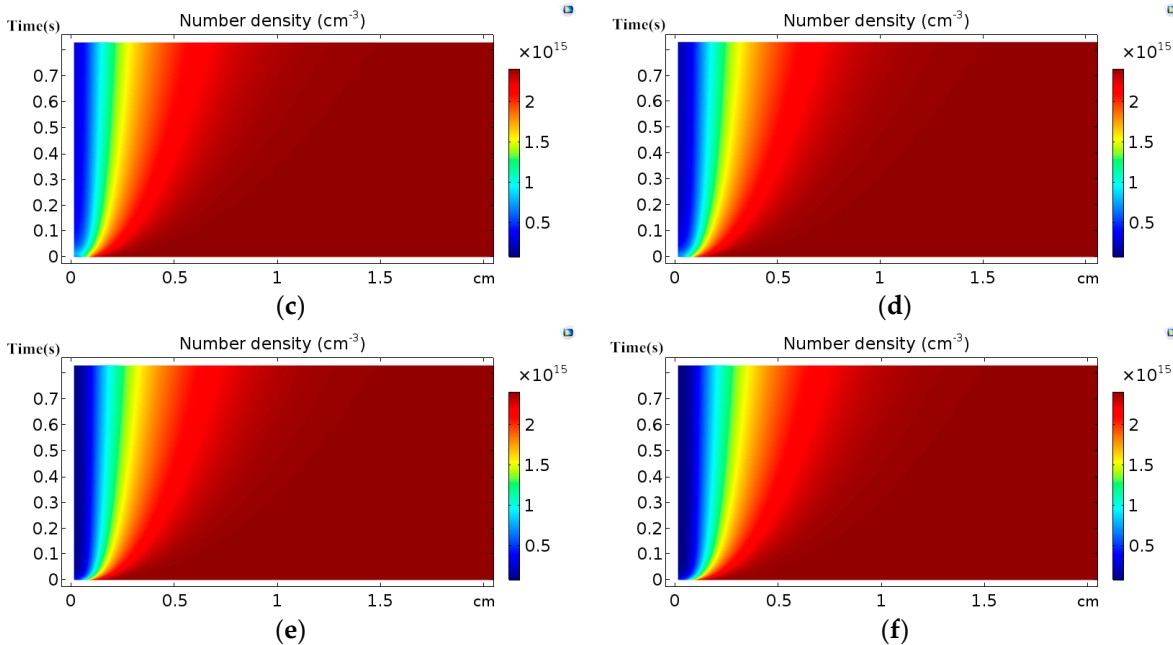

**Figure 8.** Effect of temporal and spatial evolution of number density of $SO_2$: (**a**) $-25$ KV; (**b**) $-30$ KV; (**c**) $-35$ KV; (**d**) $-40$ KV; (**e**) $-45$ KV; and (**f**) $-50$ KV.

### 3.2. Analysis of Chemical Reactions

Transient calculation was performed based on the one-dimensional fluid model and terminated to the time step of 0.83 s under the discharge voltage of $-50$ KV. The discharge gas pressure was set to one atm; the temperature was set to 300 K; the initial number densities of NO and $SO_2$ were set to $2.4 \times 10^{15}$ cm$^{-3}$ (NO concentration is about 100 ppm, $SO_2$ concentration is about 200 ppm).

Figure 9 shows the distribution of the number densities of five kinds of active particles in the solution domain at the time steps of $2.9836 \times 10^{-5}$ s, $9.1116 \times 10^{-5}$ s, $2.7826 \times 10^{-4}$ s, $8.4975 \times 10^{-4}$ s, and $2.595 \times 10^{-3}$ s. It can be seen that the five kinds of active particles are mainly concentrated in the high electric field region after discharge, and their number densities are gradually decreasing, or increasing first and then decreasing, which is due to their participation in the reactions removing NO and $SO_2$. The number densities of O and N are higher than that of the other three active species, because $N_2$ and $O_2$ are the main components of background gas. Specifically, O and O($^1$D) mainly come from the electron collision reactions of R9 and R10; N and $N_2$(A) mainly come from the reactions of R36 and R24; meanwhile, OH mainly comes from the reaction between O($^1$D) and $H_2O$, as shown in R99.

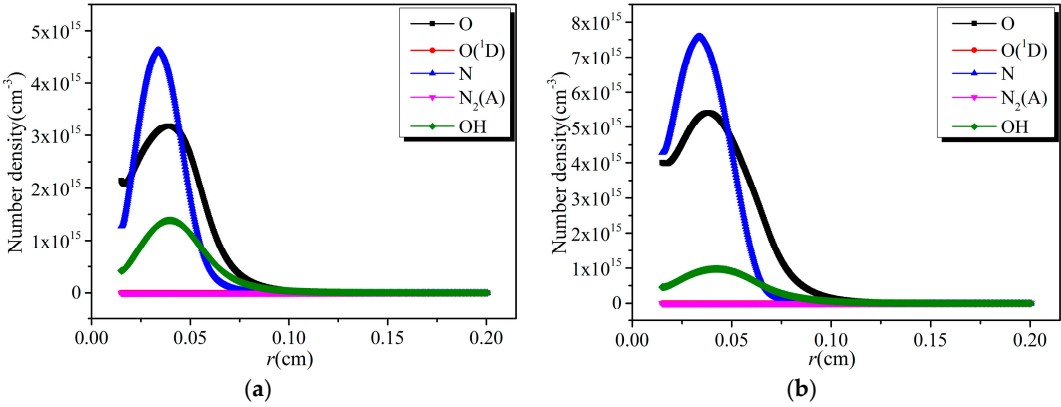

**Figure 9.** *Cont.*

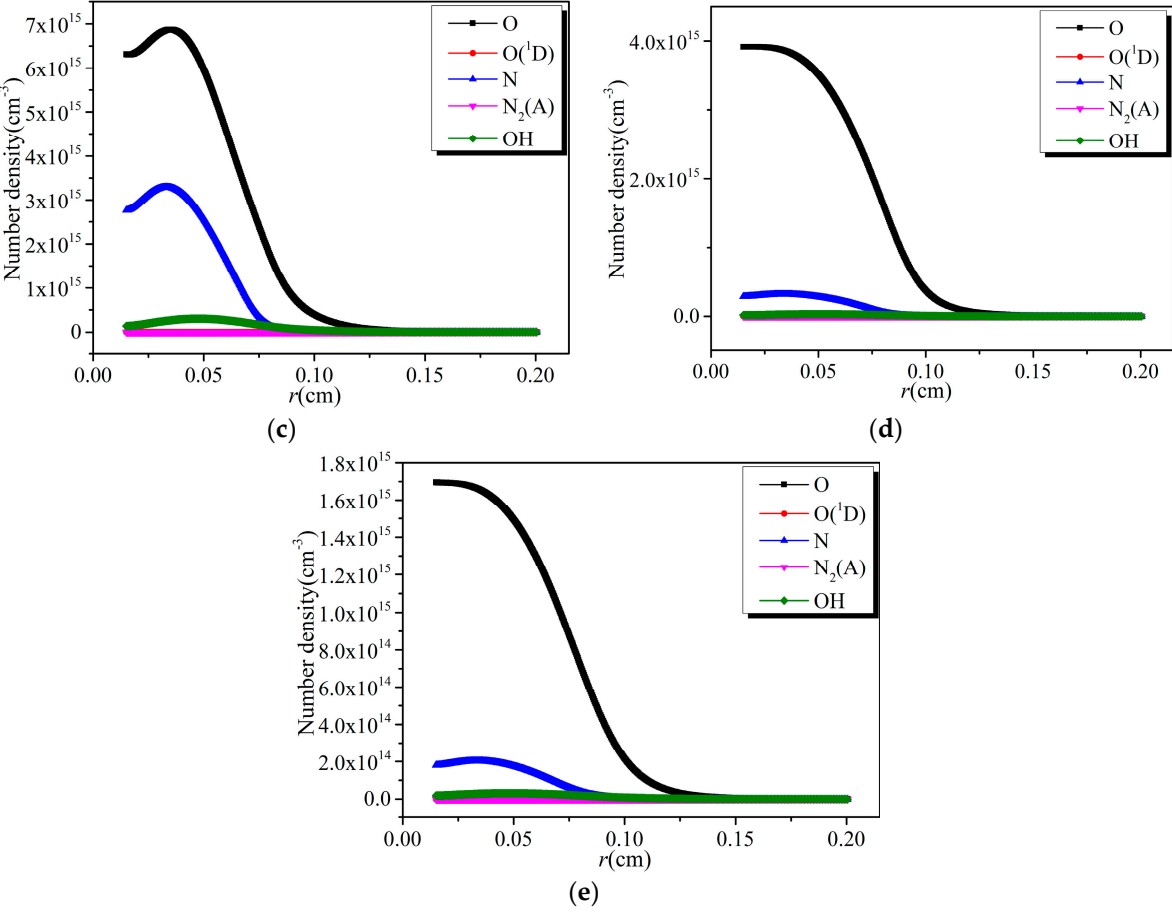

**Figure 9.** Distribution of the number densities of active particles at various time steps: (**a**) $2.9836 \times 10^{-5}$ s; (**b**) $9.1116 \times 10^{-5}$ s; (**c**) $2.7826 \times 10^{-4}$ s; (**d**) $8.4975 \times 10^{-4}$ s; and (**e**) $2.595 \times 10^{-3}$ s.

Figure 10 shows the distribution of the number densities of NO, $SO_2$, and their main oxidation and reduction products by NTP at the time steps of $2.9836 \times 10^{-5}$ s, $9.1116 \times 10^{-5}$ s, $2.7826 \times 10^{-4}$ s, $8.4975 \times 10^{-4}$ s, and and 0.002595 s. $NO_2$ mainly comes from the oxidation of NO by oxidizing particles such as O and $O_3$. However, there is also a reduction reaction between O and $NO_2$, as shown in R75. Therefore, the number density of $NO_2$ presents a trend of first rising and then decreasing. $NO_3$ and $N_2O_5$ come from the further oxidation of $NO_2$ and the reactions between oxidation products, as shown in R74, R76, and R79. Their number densities increase with time and are one order of magnitude smaller than $NO_2$. $HNO_2$ and $HNO_3$ mainly come from the oxidation of NO and $NO_2$ by OH, as shown in R113 and R114. The sum of the number densities of $HNO_2$ and $HNO_3$ presents a trend of first rising and then decreasing because of the existence of R115. The increase of the number density of $N_2O$, which is a kind of greenhouse gas, mainly comes from R83 and R90, while the decrease of it mainly comes from R93, R110, and R111. $SO_3$ mainly comes from the oxidation of $SO_2$ by O, as shown in R120, while there is also a reduction reaction between O and $SO_3$, as shown in R121. The number density of $SO_3$ decreases with time because the rate coefficient of R121 is much larger than that of R120. $HSO_3$ and $H_2SO_4$ mainly come from the oxidation of $SO_2$ and $HSO_3$ by OH, as shown in R118 and R119. It can be seen that the sum of the number densities of $HSO_3$ and $H_2SO_4$ is several orders of magnitude larger than that of $SO_3$, so the oxidation by OH is the main path of $SO_2$ removal.

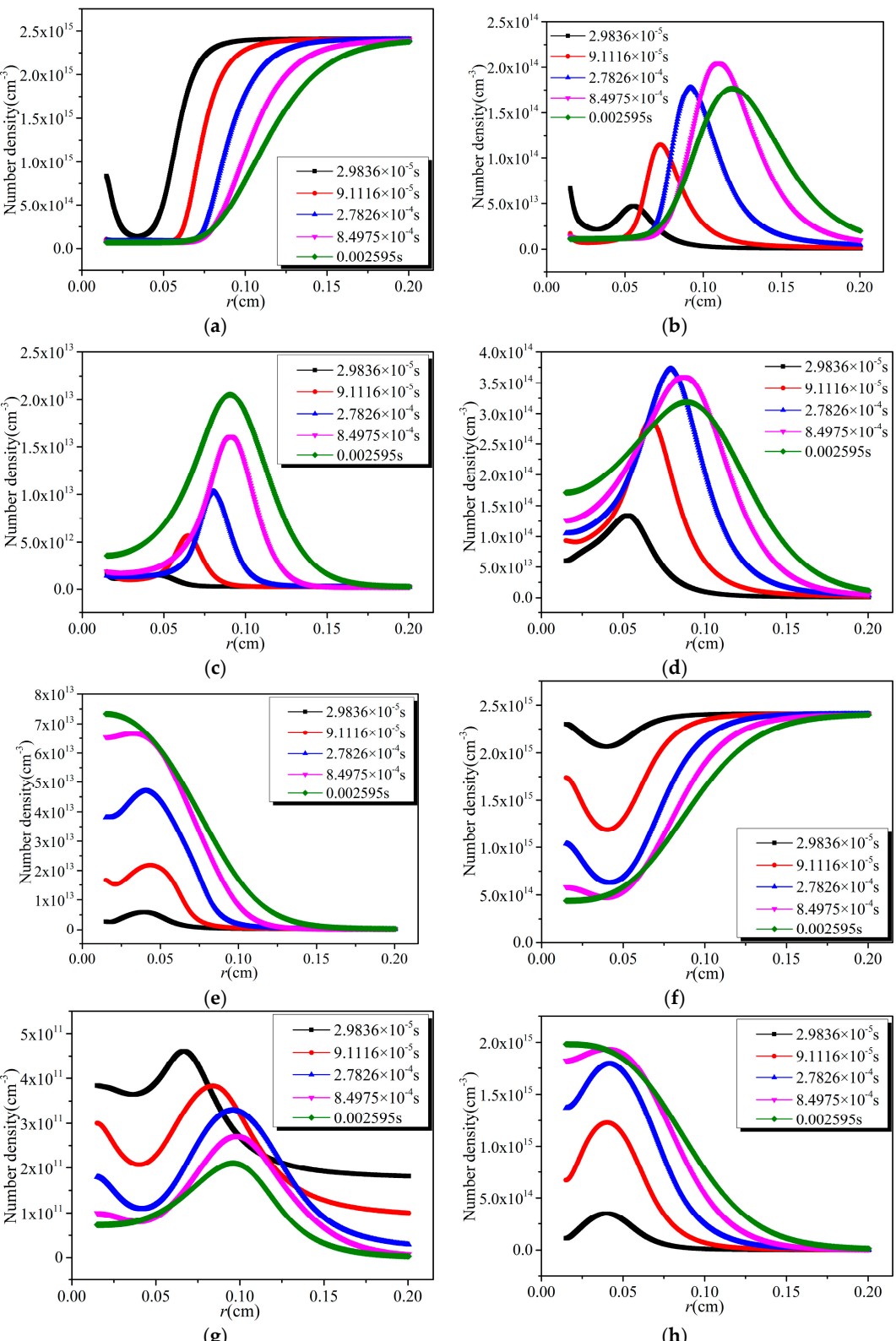

**Figure 10.** Distribution of the number densities of NO, SO$_2$, and their main oxidation and reduction products by non-thermal plasma (NTP) at various time steps: (**a**) NO; (**b**) NO$_2$; (**c**) NO$_3$+N$_2$O$_5$; (**d**) HNO$_2$+HNO$_3$; (**e**) N$_2$O; (**f**) SO$_2$; (**g**) SO$_3$; and (**h**) HSO$_3$+H$_2$SO$_4$.

The reduction reaction with N is another important removal route for NO in NTP besides oxidation by O, $O_3$, and OH [28], as shown in R81. The variable $\eta$ is defined as the percentage of NO removed by oxidation in the total amount of NO removed, as shown in Equation (25):

$$\eta = \frac{N_{NO_2} + N_{NO_3} + 2 \times N_{N_2O_5} + 2 \times N_{N_2O} + N_{HNO_2} + N_{HNO_3}}{N_{NO}^0 - N_{NO}} \tag{25}$$

where $N_{NO}^0$ is the initial number density of NO.

Figure 11 shows the distribution of $\eta$ in the effective removal area of NO at the time steps of $2.9836 \times 10^{-5}$ s, $9.1116 \times 10^{-5}$ s, $2.7826 \times 10^{-4}$ s, $8.4975 \times 10^{-4}$ s, and $2.595 \times 10^{-3}$ s. It can be seen that the value of $\eta$ increases with the increase of the distance from the discharge electrode, which indicates that the removal of NO mainly depends on the reduction process in the region near the discharge electrode, while it mainly depends on the oxidation process in the region far away from the cathode. This phenomenon is related to the distribution of active particles. As can be seen from Figure 9, the peak position of the number density of N is closer to the discharge electrode than that of O and OH. Therefore, the location of the reduction process of NO is closer to the discharge electrode than that of the oxidation of NO.

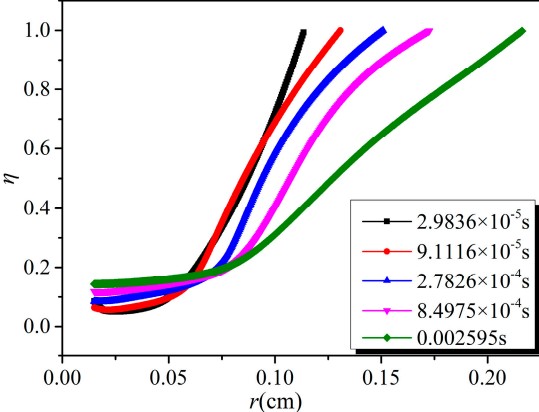

**Figure 11.** Distribution of $\eta$ in the effective removal area of NO at various time steps.

## 4. Conclusions

In this paper, a one-dimensional fluid model of the removal of NO and $SO_2$ by negative DC corona discharge based on the coaxial wire-cylindrical electrodes configuration was established and solved by the finite element method. The coupled electron collision kinetic model in the fluid model contains 64 elementary reactions and the chemical kinetic model contains 117 elementary reactions. Transient calculation was performed and terminated to the time step of 0.83 s. The background gas system is composed of $O_2$, $N_2$, $H_2O$, and $CO_2$ with a volume ratio of 0.033:0.741:0.083:0.143. The main conclusions obtained are as follows.

(1) The reduced electric field decreases sharply from the edge of the discharge electrode, forming a narrow high electric field region about one mm away from the cathode. The high electric field region is also the high electron energy region, where the electron energy is close to each other under different discharge voltages.

(2) The reaction rates of electron collision reactions are higher in the high electric field region and tend to zero gradually outside this region. There is a peak value in the spatial distribution of various reaction rates, whose position is determined by the location of the electron energy threshold corresponding to various reactions.

(3) The distribution of the number densities of active particles between two electrodes has the same trend as that of the corresponding electron collision reaction rates.

(4)　The effective removal area of NO and SO$_2$ increases with time, and the removal efficiency decreases with the increase of distance from the discharge electrode under the same discharge voltage. With the increase of discharge voltage, the effective removal area of NO and SO$_2$, as well as the removal efficiency at the same position, increase.

(5)　The removal of NO mainly depends on the reduction process in the region near the discharge electrode, while it mainly depends on the oxidation process in the region far away from the cathode. The sum of the number densities of HSO$_3$ and H$_2$SO$_4$ is several orders of magnitude larger than that of SO$_3$, which indicates that the oxidation by OH is the main path of SO$_2$ removal.

**Author Contributions:** Conceptualization, L.Y.; Formal analysis, C.Z.; Investigation, C.Z.; Methodology, C.Z.; Supervision, L.Y.; Writing—original draft, C.Z.

**Funding:** This research received no external funding.

**Conflicts of Interest:** The authors declare no conflict of interest.

## Appendix A

**Table A1.** Electron collision kinetic model.

| Number | Reaction Equation | Number | Reaction Equation |
|--------|-------------------|--------|-------------------|
| R1 | $e + O_2 \rightarrow e + O_2$ | R2 | $e + O_2 \rightarrow O_2^-$ |
| R3 | $e + O_2 \rightarrow O + O^-$ | R4 | $e + O_2 \rightarrow e + O_2(ROT)$ |
| R5 | $e + O_2 \rightarrow e + O_2(VSUM)$ | R6 | $e + O_2 \rightarrow e + O_2(a^1\Delta_g)$ |
| R7 | $e + O_2 \rightarrow e + O_2(b^1\sum g^+)$ | R8 | $e + O_2 \rightarrow e + O_2(4.5)$ |
| R9 | $e + O_2 \rightarrow e + O + O$ | R10 | $e + O_2 \rightarrow e + O + O(^1D)$ |
| R11 | $e + O_2 \rightarrow e + e + O_2^+$ | R12 | $e + O_2 \rightarrow e + e + O + O^+$ |
| R13 | $e + N_2 \rightarrow e + N_2$ | R14 | $e + N_2 \rightarrow e + N_2^*(0.02eV)$ |
| R15 | $e + N_2 \rightarrow e + N_2(V1) (0.29eV)$ | R16 | $e + N_2 \rightarrow e + N_2(V1) (0.291eV)$ |
| R17 | $e + N_2 \rightarrow e + N_2(V2) (0.59eV)$ | R18 | $e + N_2 \rightarrow e + N_2(V3) (0.88eV)$ |
| R19 | $e + N_2 \rightarrow e + N_2(V4) (1.17eV)$ | R20 | $e + N_2 \rightarrow e + N_2(V5) (1.47eV)$ |
| R21 | $e + N_2 \rightarrow e + N_2(V6) (1.76eV)$ | R22 | $e + N_2 \rightarrow e + N_2(V7) (2.06eV)$ |
| R23 | $e + N_2 \rightarrow e + N_2(V8) (2.35eV)$ | R24 | $e + N_2 \rightarrow e + N_2(A)$ |
| R25 | $e + N_2 \rightarrow e + N_2^*(7eV)$ | R26 | $e + N_2 \rightarrow e + N_2^*(7.35eV)$ |
| R27 | $e + N_2 \rightarrow e + N_2^*(7.36eV)$ | R28 | $e + N_2 \rightarrow e + N_2^*(7.8eV)$ |
| R29 | $e + N_2 \rightarrow e + N_2^*(8.16eV)$ | R30 | $e + N_2 \rightarrow e + N_2^*(8.4eV)$ |
| R31 | $e + N_2 \rightarrow e + N_2^*(8.55eV)$ | R32 | $e + N_2 \rightarrow e + N_2^*(8.89eV)$ |
| R33 | $e + N_2 \rightarrow e + N_2^*(11.03eV)$ | R34 | $e + N_2 \rightarrow e + N_2^*(11.88eV)$ |
| R35 | $e + N_2 \rightarrow e + N_2^*(12.25eV)$ | R36 | $e + N_2 \rightarrow e + N + N(13eV)$ |
| R37 | $e + N_2 \rightarrow e + e + N_2^+$ | R38 | $e + H_2O \rightarrow e + H_2O$ |
| R39 | $e + H_2O \rightarrow H_2 + O^-$ | R40 | $e + H_2O \rightarrow OH + H^-$ |
| R41 | $e + H_2O \rightarrow e + H_2O(ROT)$ | R42 | $e + H_2O \rightarrow e + H_2O(010)$ |
| R43 | $e + H_2O \rightarrow e + H_2O(101)$ | R44 | $e + H_2O \rightarrow e + H + OH$ |
| R45 | $e + H_2O \rightarrow e + e + H_2O^+$ | R46 | $e + CO_2 \rightarrow e + CO_2$ |
| R47 | $e + CO_2 \rightarrow CO + O^-$ | R48 | $e + CO_2 \rightarrow e + CO_2(010)$ |
| R49 | $e + CO_2 \rightarrow e + CO_2(100)$ | R50 | $e + CO_2 \rightarrow e + CO_2(110)$ |
| R51 | $e + CO_2 \rightarrow e + CO_2(001)$ | R52 | $e + CO_2 \rightarrow e + CO_2(200)$ |
| R53 | $e + CO_2 \rightarrow e + CO_2(210)$ | R54 | $e + CO_2 \rightarrow e + CO_2(300)$ |
| R55 | $e + CO_2 \rightarrow e+ CO_2/1010$ | R56 | $e + CO_2 \rightarrow e + CO_2(7)$ |
| R57 | $e + CO_2 \rightarrow e + CO_2(8)$ | R58 | $e + CO_2 \rightarrow e + CO_2(9)$ |
| R59 | $e + CO_2 \rightarrow e + CO_2/11.1$ | R60 | $e + CO_2 \rightarrow e + CO_2/11.9$ |
| R61 | $e + CO_2 \rightarrow e + CO_2/12.4$ | R62 | $e + CO_2 \rightarrow e + CO_2/17.3$ |
| R63 | $e + CO_2 \rightarrow e + CO_2/18.1$ | R64 | $e + CO_2 \rightarrow e + e + CO_2^+$ |

## Appendix B

The rate coefficients of all the reactions in Table A2 are the values at 300 K; the rate coefficients of the two-body reactions are in cm$^3$s$^{-1}$, and the rate coefficients of the three-body reactions are in cm$^6$s$^{-1}$. M represents the third body in the collision reactions, whose number density [M] can be set to $2.422 \times 10^{19}$ cm$^{-3}$ at 300 K [16]. The rate coefficient of R126 is derived from the estimated value in the corresponding literature.

**Table A2.** Chemical kinetic model.

| Number | Reaction Equation | Rate Coefficient | Reference |
|---|---|---|---|
| R65 | $O_2 + O_2(a^1\Delta_g) \rightarrow 2O_2$ | $2.2 \times 10^{-18}$ | [15] |
| R66 | $O_2(a^1\Delta_g) + O_2(a^1\Delta_g) \rightarrow 2O_2$ | $2.0 \times 10^{-17}$ | [15] |
| R67 | $O + O_2 \rightarrow O_3$ | $1.0 \times 10^{-14}$ | [15] |
| R68 | $N_2 + O_3 \rightarrow N_2 + O_2 + O$ | $2.0 \times 10^{-26}$ | [15] |
| R69 | $O + O + N_2 \rightarrow O_2 + N_2$ | $8.0 \times 10^{-33}$ | [15] |
| R70 | $O + O_3 \rightarrow 2O_2$ | $1.0 \times 10^{-14}$ | [15] |
| R71 | $OH + OH \rightarrow H_2O + O$ | $2.0 \times 10^{-12}$ | [15] |
| R72 | $OH + H + N_2 \rightarrow H_2O + N_2$ | $5.0 \times 10^{-31}$ | [15] |
| R73 | $O + NO + N_2 \rightarrow NO_2 + N_2$ | $9.0 \times 10^{-32}$ | [15] |
| R74 | $O + NO_2 + N_2 \rightarrow NO_3 + N_2$ | $9.0 \times 10^{-32}$ | [15] |
| R75 | $O + NO_2 \rightarrow NO + O_2$ | $1.0 \times 10^{-11}$ | [15] |
| R76 | $NO_3 + NO_2 + N_2 \rightarrow N_2O_5 + N_2$ | $2.2 \times 10^{-30}$ | [15] |
| R77 | $NO_3 + NO \rightarrow 2NO_2$ | $3.0 \times 10^{-11}$ | [15] |
| R78 | $NO_3 + O \rightarrow NO_2 + O_2$ | $1.7 \times 10^{-11}$ | [15] |
| R79 | $NO_2 + O_3 \rightarrow NO_3 + O_2$ | $3.0 \times 10^{-17}$ | [15] |
| R80 | $NO + O_3 \rightarrow NO_2 + O_2$ | $1.8 \times 10^{-14}$ | [15] |
| R81 | $N + NO \rightarrow N_2 + O$ | $5.9 \times 10^{-11}$ | [15] |
| R82 | $N + NO_2 \rightarrow 2NO$ | $9.0 \times 10^{-12}$ | [15] |
| R83 | $N + NO_2 \rightarrow N_2O + O$ | $3.0 \times 10^{-12}$ | [15] |
| R84 | $N + N + N_2 \rightarrow N_2 + N_2$ | $4.0 \times 10^{-33}$ | [15] |
| R85 | $N + O + N_2 \rightarrow NO + N_2$ | $1.0 \times 10^{-32}$ | [15] |
| R86 | $N + O_2 \rightarrow NO + O$ | $1.0 \times 10^{-16}$ | [15] |
| R87 | $N + O_3 \rightarrow NO + O_2$ | $1.0 \times 10^{-16}$ | [15] |
| R88 | $N_2(A) + O_2 \rightarrow O_2(a^1\Delta_g) + N_2$ | $1.0 \times 10^{-12}$ | [15] |
| R89 | $N_2(A) + O_2 \rightarrow N_2 + O + O$ | $2.5 \times 10^{-12}$ | [15] |
| R90 | $N_2(A) + O_2 \rightarrow N_2O + O$ | $7.8 \times 10^{-14}$ | [15] |
| R91 | $N_2(A) + O \rightarrow NO + N$ | $7.0 \times 10^{-12}$ | [15] |
| R92 | $N_2(A) + O \rightarrow N_2 + O$ | $2.0 \times 10^{-11}$ | [15] |
| R93 | $N_2(A) + N_2O \rightarrow N_2 + N + NO$ | $1.0 \times 10^{-11}$ | [15] |
| R94 | $N_2(A) + N_2O \rightarrow 2N_2 + O$ | $1.0 \times 10^{-11}$ | [15] |
| R95 | $N_2(A) + N_2(A) \rightarrow 2N_2$ | $2.0 \times 10^{-12}$ | [15] |
| R96 | $N_2(A) + N_2 \rightarrow 2N_2$ | $3.0 \times 10^{-16}$ | [15] |
| R97 | $N_2(A) + NO \rightarrow N_2 + NO$ | $7.0 \times 10^{-11}$ | [15] |
| R98 | $N_2(A) + NO_2 \rightarrow N_2 + NO + O$ | $1.0 \times 10^{-12}$ | [15] |
| R99 | $O(^1D) + H_2O \rightarrow 2OH$ | $2.2 \times 10^{-10}$ | [15] |
| R100 | $O(^1D) + H_2O \rightarrow H_2O + O$ | $1.2 \times 10^{-11}$ | [15] |
| R101 | $O(^1D) + O_3 \rightarrow 2O + O_2$ | $1.2 \times 10^{-10}$ | [15] |
| R102 | $O(^1D) + O_3 \rightarrow 2O_2$ | $1.2 \times 10^{-10}$ | [15] |
| R103 | $O(^1D) + NO \rightarrow N + O_2$ | $1.7 \times 10^{-10}$ | [15] |
| R104 | $O(^1D) + NO_2 \rightarrow NO + O_2$ | $1.4 \times 10^{-10}$ | [15] |
| R105 | $O(^1D) + N_2 \rightarrow O + N_2$ | $2.6 \times 10^{-11}$ | [15] |
| R106 | $O(^1D) + O_2 \rightarrow O + O_2(a^1\Delta_g)$ | $3.4 \times 10^{-11}$ | [15] |
| R107 | $O(^1D) + O_2 \rightarrow O + O_2$ | $6.3 \times 10^{-12}$ | [15] |
| R108 | $O(^1D) + N_2 \rightarrow O + N_2$ | $1.8 \times 10^{-11}$ | [15] |
| R109 | $O(^1D) + O + N_2 \rightarrow O_2 + N_2$ | $9.9 \times 10^{-33}$ | [15] |
| R110 | $O(^1D) + N_2O \rightarrow 2NO$ | $6.7 \times 10^{-11}$ | [15] |
| R111 | $O(^1D) + N_2O \rightarrow N_2 + O_2$ | $4.9 \times 10^{-11}$ | [15] |
| R112 | $O(^1D) + N_2 + N_2 \rightarrow N_2O + N_2$ | $3.5 \times 10^{-37}$ | [15] |
| R113 | $NO_2 + OH \rightarrow HNO_3$ | $1.0 \times 10^{-11}$ | [15] |
| R114 | $NO + OH \rightarrow HNO_2$ | $6.6 \times 10^{-12}$ | [15] |
| R115 | $HNO_2 + OH \rightarrow NO_2 + H_2O$ | $0.5 \times 10^{-11}$ | [15] |
| R116 | $HNO_2 + HNO_2 \rightarrow NO + NO_2 + H_2O$ | $1.0 \times 10^{-20}$ | [15] |
| R117 | $HNO_2 + O \rightarrow NO_2 + OH$ | $3.0 \times 10^{-15}$ | [15] |
| R118 | $OH + SO_2 \rightarrow HSO_3$ | $7.5 \times 10^{-12}$ | [15] |
| R119 | $OH + HSO_3 \rightarrow H_2SO_4$ | $1.0 \times 10^{-12}$ | [15] |

**Table A2.** *Cont.*

| Number | Reaction Equation | Rate Coefficient | Reference |
|--------|-------------------|------------------|-----------|
| R120 | $O + SO_2 + N_2 \rightarrow SO_3 + N_2$ | $1.4 \times 10^{-33}$ | [15] |
| R121 | $O + SO_3 + N_2 \rightarrow SO_2 + O_2 + N_2$ | $8.0 \times 10^{-30}$ | [15] |
| R122 | $SO_3 + H_2O \rightarrow H_2SO_4$ | $6.0 \times 10^{-15}$ | [15] |
| R123 | $CO + O + N_2 \rightarrow CO_2 + N_2$ | $4.7 \times 10^{-36}$ | [15] |
| R124 | $CO + OH \rightarrow CO_2 + H$ | $1.5 \times 10^{-13}$ | [15] |
| R125 | $NO_3 + CO \rightarrow NO_2 + CO_2$ | $3.2 \times 10^{-16}$ | [15] |
| R126 | $N + CO_2 \rightarrow NO + CO$ | $1.0 \times 10^{-14}$ | [15] |
| R127 | $N_2^+ + e \rightarrow N_2$ | $4.0 \times 10^{-12} + 6.0 \times 10^{-27}$ [M] | [16] |
| R128 | $N_2^+ + O_2^- \rightarrow N_2 + O_2$ | $1.6 \times 10^{-7} + 3.0 \times 10^{-25}$ [M] | [16] |
| R129 | $N_2^+ + O^- \rightarrow N_2 + O$ | $4.0 \times 10^{-7} + 3.0 \times 10^{-25}$ [M] | [16] |
| R130 | $N_2^+ + NO^- \rightarrow N_2 + NO$ | $4.0 \times 10^{-7} + 3.0 \times 10^{-25}$ [M] | [16] |
| R131 | $N_2^+ + NO_2^- \rightarrow NO_2 + N_2$ | $4.0 \times 10^{-7} + 3.0 \times 10^{-25}$ [M] | [16] |
| R132 | $O_2^+ + e \rightarrow O + O(^1D)$ | $2.1 \times 10^{-7}$ | [16] |
| R133 | $O_2^+ + e \rightarrow O_2$ | $4.0 \times 10^{-12} + 6.0 \times 10^{-27}$ [M] | [16] |
| R134 | $O_2^+ + O^- \rightarrow O_2 + O$ | $9.6 \times 10^{-8} + 3.0 \times 10^{-25}$ [M] | [16] |
| R135 | $O_2^+ + O_2^- \rightarrow 2O_2$ | $4.2 \times 10^{-7} + 3.0 \times 10^{-25}$[M] | [16] |
| R136 | $H^- + O_2^+ \rightarrow H + O_2$ | $2.0 \times 10^{-7}$ | [15] |
| R137 | $O_2^+ + NO^- \rightarrow NO + O_2$ | $4.0 \times 10^{-7} + 3.0 \times 10^{-25}$ [M] | [16] |
| R138 | $O_2^+ + NO_2^- \rightarrow NO_2 + O_2$ | $4.1 \times 10^{-7} + 3.0 \times 10^{-25}$ [M] | [16] |
| R139 | $O^+ + e \rightarrow O$ | $4.0 \times 10^{-12} + 6.0 \times 10^{-27}$ [M] | [16] |
| R140 | $O^+ + O^- \rightarrow 2O$ | $2.7 \times 10^{-7} + 3.0 \times 10^{-25}$ [M] | [16] |
| R141 | $O^+ + O_2^- \rightarrow O + O_2$ | $4.0 \times 10^{-7} + 3.0 \times 10^{-25}$ [M] | [16] |
| R142 | $O^+ + NO^- \rightarrow NO + O$ | $4.0 \times 10^{-7} + 3.0 \times 10^{-25}$ [M] | [16] |
| R143 | $O^+ + NO_2^- \rightarrow NO + O_2$ | $4.0 \times 10^{-7} + 3.0 \times 10^{-25}$ [M] | [16] |
| R144 | $H_2O^+ + e \rightarrow OH + H$ | $3.8 \times 10^{-7}$ | [16] |
| R145 | $H_2O^+ + e \rightarrow H_2 + O$ | $1.4 \times 10^{-7}$ | [16] |
| R146 | $H_2O^+ + e \rightarrow 2H + O$ | $1.7 \times 10^{-7}$ | [16] |
| R147 | $H_2O^+ + e + M \rightarrow H_2O + M$ | $6.0 \times 10^{-27}$ | [16] |
| R148 | $H_2O^+ + O^- \rightarrow H_2O + O$ | $4.0 \times 10^{-7} + 3.0 \times 10^{-25}$ [M] | [16] |
| R149 | $H_2O^+ + O_2^- \rightarrow H_2O + O_2$ | $4.0 \times 10^{-7} + 3.0 \times 10^{-25}$ [M] | [16] |
| R150 | $H_2O^+ + NO^- \rightarrow NO + H_2O$ | $4.0 \times 10^{-7} + 3.0 \times 10^{-25}$ [M] | [16] |
| R151 | $H_2O^+ + NO_2^- \rightarrow NO_2 + H_2O$ | $4.0 \times 10^{-7} + 3.0 \times 10^{-25}$ [M] | [16] |
| R152 | $CO_2^+ + e \rightarrow CO + O$ | $4.0 \times 10^{-7}$ | [16] |
| R153 | $CO_2^+ + e + M \rightarrow CO_2 + M$ | $6.0 \times 10^{-27}$ | [16] |
| R154 | $CO_2^+ + O^- \rightarrow CO_2 + O$ | $4.0 \times 10^{-7} + 3.0 \times 10^{-25}$ [M] | [16] |
| R155 | $CO_2^+ + O_2^- \rightarrow CO_2 + O_2$ | $4.0 \times 10^{-7} + 3.0 \times 10^{-25}$ [M] | [16] |
| R156 | $CO_2^+ + NO^- \rightarrow CO_2 + NO$ | $4.0 \times 10^{-7} + 3.0 \times 10^{-25}$ [M] | [16] |
| R157 | $CO_2^+ + NO_2^- \rightarrow CO_2 + NO_2$ | $4.0 \times 10^{-7} + 3.0 \times 10^{-25}$ [M] | [16] |
| R158 | $NO^+ + e \rightarrow NO$ | $4.0 \times 10^{-12} + 6.0 \times 10^{-27}$ [M] | [16] |
| R159 | $NO^+ + e + M \rightarrow N + O + M$ | $1.0 \times 10^{-27}$ | [16] |
| R160 | $NO^+ + O^- \rightarrow O + NO$ | $4.9 \times 10^{-7} + 3.0 \times 10^{-25}$ [M] | [16] |
| R161 | $NO^+ + O_2^- \rightarrow NO + O_2$ | $4.0 \times 10^{-7} + 3.0 \times 10^{-25}$ [M] | [16] |
| R162 | $NO^+ + NO^- \rightarrow O_2 + N_2$ | $4.0 \times 10^{-7} + 3.0 \times 10^{-25}$ [M] | [16] |
| R163 | $NO^+ + NO_2^- \rightarrow NO_2 + N + O$ | $1.0 \times 10^{-7}$ | [16] |
| R164 | $NO^+ + NO_2^- \rightarrow NO + NO_2$ | $3.5 \times 10^{-7} + 3.0 \times 10^{-25}$ [M] | [16] |
| R165 | $NO_2^+ + e \rightarrow NO + O$ | $3.0 \times 10^{-7}$ | [16] |
| R166 | $NO_2^+ + e + M \rightarrow NO_2 + M$ | $6.0 \times 10^{-27}$ | [16] |
| R167 | $NO_2^+ + O^- \rightarrow NO + O_2$ | $4.0 \times 10^{-7} + 3.0 \times 10^{-25}$ [M] | [16] |
| R168 | $NO_2^+ + O_2^- \rightarrow NO_2 + O_2$ | $4.0 \times 10^{-7} + 3.0 \times 10^{-25}$ [M] | [16] |
| R169 | $NO_2^+ + NO^- \rightarrow N_2O + O_2$ | $4.0 \times 10^{-7} + 3.0 \times 10^{-25}$ [M] | [16] |
| R170 | $NO_2^+ + NO_2^- \rightarrow 2O_2 + N_2$ | $4.0 \times 10^{-7} + 3.0 \times 10^{-25}$ [M] | [16] |
| R171 | $O_2^+ + NO \rightarrow NO^+ + O_2$ | $3.5 \times 10^{-10}$ | [16] |
| R172 | $O_2^+ + NO_2 \rightarrow NO_2^+ + O_2$ | $6.0 \times 10^{-10}$ | [16] |
| R173 | $O_2^+ + NO_2 \rightarrow NO^+ + O_3$ | $1.0 \times 10^{-11}$ | [15] |
| R174 | $O_2^+ + N \rightarrow NO^+ + O$ | $1.8 \times 10^{-10}$ | [16] |
| R175 | $O^- + O_2 \rightarrow O_2^- + O$ | $1.0 \times 10^{-10}$ | [15] |
| R176 | $O^- + NO \rightarrow e + NO_2$ | $3.1 \times 10^{-10}$ | [16] |
| R177 | $O^- + SO_2 \rightarrow SO_3 + e$ | $2.0 \times 10^{-9}$ | [15] |
| R178 | $e + 2O_2 \rightarrow O_2^- + O_2(a^1\Delta_g)$ | $3.3 \times 10^{-39}$ | [16] |
| R179 | $e + NO + M \rightarrow NO^- + M$ | $8.0 \times 10^{-31}$ | [16] |
| R180 | $e + NO_2 + M \rightarrow NO_2^- + M$ | $1.5 \times 10^{-30}$ | [16] |
| R181 | $e + HNO_3 \rightarrow NO_2^- + OH$ | $5.0 \times 10^{-8}$ | [16] |

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
