# Peer review of "One-Dimensional Simulation of Synergistic Desulfurization and Denitrification Processes for Electrostatic Precipitators Based on a Fluid-Chemical Reaction Hybrid Model"

_energies, doi:10.3390/en11123249_

Round 1

Reviewer 1 Report

The authors present a simplified chemical kinetic fluid model for a co-axial DC discharge to study the removal of NOx and SOx from flue gas. The manuscript is well written, with a clear theory section to support the model results. I do have a few essential points that need to be addressed. The really important issues are with providing a proper comparison of the presented model to experimental data, as I outline in several points below.

Major:

- Is the model 1D or 1D-axisymmetric? This is not made clear in Section 2.4, even though this is an essential distinction. The results for E/n0 suggest it is axisymmetric, since the (reduced) E-field likely has an ln(r) type decay in Figure 3. Are all transport equations also solved axisymmetrically (i.e. is there effectively more volume as r increases)? My guess is yes, but it is not clear from the text.

- It would be insightful to show the electron density somewhere and make mention of the dominant ion species. Is the ionization degree (as function of distance from the wire electrode) consistent with what is expected for a DC negative Corona? Comparison to experimental literature on electron density in corona would also be valuable to assess the accuracy of the model.

- In the same category: can a current or power be extracted from the model? These numbers could again be compared to experimental literature to validate the accuracy of the model.

- And again: Discussion on how the model would compare to real devices is in order: I would assume that a device including 'depth' (i.e. having a z-axis) would have at its output the composition calculated here at high values of time. What length of device would correspond to t = 2.6 ms at a practical flow rate? What does the model predict in terms of fractional NO and SO removal? How does this correspond to actual devices? This would really help to show your kinetic model is sufficient to describe real devices and improve the impact of the work.

Minor:

- Equation (18) shows the loss of electron energy to the surface. There are, however, also secondary electrons being considered, but no boundary condition for their energy is explicitly provided. This energy is part of the model, as it is mentioned in line 245, so I believe this should be included as part of Equation (18) or elsewhere.

- In Figure 7 onwards, the y-axis is labeled only as 's'. This should be 'Time (s)', I believe.

- Another small thing: the times in line 354 and subsequent figures could be rounded to 2 significant digits to improve clarity, without loss of relevant accuracy.

Author Response

Dear Reviewer,

Thank you very much for your comments on our article. We have responded to your comments in the attached PDF file. Please check it.

Best regards.

Reviewer 2 Report

The authors present interesting results on the usage of Corona plasmas for NOx and SO2 removal. The results and conclusions are of interest for practical applications and shall be well regarded by the communicty utilizing Corona treatments for the purification of exhaust gases.

The introduction gives a very brief overview of the topic, sufficient for the scope and main audience.

The model description and the deductions of formulae appear solid and reliable. However, improvements to the references are indicated:
[17], [18], [19] and [20]: All four are not accessible solely via their title. Please provide permalink or unique identifyer to make them accessible. Otherwise, these should be replaced by more suited references.
[19]: Although findable by title, it would be advisable to provide either the permanent link (www.vssd.nl/hlf/d004.htm) or ISBN (978-90-71301-58-2)

Author Response

(The authors gave the same response as above.)

Reviewer 3 Report

Chao Zhang and Lixin Yang investigated the synergistic desulfurization and denitrification process for the electrostatic precipitators based on a fluid-chemical reaction model using the 1-D simulation. The plasma dynamics of the desulfurization and denitrification is an important problem for the NTP pollutant control technologies. The authors used a 1-dimensional fluid model to simulate the desulfurization and denitrification of flue gas by negative DC corona discharge based on the traditional 0-dimensional chemical kinetic model. It is found that the removal efficiency of NO and SO2 is remarkable in the region with a small radius around the high voltage electrode. Such effective purification area expands with the increase of the discharge voltage. And, the removal of SO2 is mainly dependent on the oxidation by OH. In contrast, there are different removal pathways for NO at different positions in the removal region. 

This is a very strong work with adequately information, data, as well as discussion. I recommend an acceptance with the current version. 

Author Response

(The authors gave the same response as above.)
